# The Challenge of Reaching Undocumented Migrants with COVID-19 Vaccination

**DOI:** 10.3390/ijerph19169973

**Published:** 2022-08-12

**Authors:** Stephen A. Matlin, Alyna C. Smith, Jessica Merone, Michele LeVoy, Jalpa Shah, Frank Vanbiervliet, Stéphanie Vandentorren, Joanna Vearey, Luciano Saso

**Affiliations:** 1Institute of Global Health Innovation, Imperial College London, South Kensington, London SW7 2AZ, UK; 2Rue du Congrès/Congresstraat 37-41, P.O. Box 5, 1000 Brussels, Belgium; 3Human Rights Center, University of Padova, Via 8 Febbraio, 2, 35122 Padova, Italy; 4Santé Publique France, 12 rue du Val d’Osne, CEDEX, 94415 Saint-Maurice, France; 5Bruss’help, 1000 Brussels, Belgium; 6INSERM UMR 1219-Bordeaux Population Health, University of Bordeaux, 33000 Bordeaux, France; 7African Centre for Migration & Society (ACMS), University of the Witwatersrand, Johannesburg 2000, South Africa; 8Department of Physiology and Pharmacology, Sapienza University of Rome, 00185 Rome, Italy

**Keywords:** undocumented migrants, COVID-19 vaccination, supply- and demand-side access barriers, equitable access

## Abstract

Access to vaccination against a health threat such as that presented by the COVID-19 pandemic is an imperative driven, in principle, by at least three compelling factors: (1) the right to health of all people, irrespective of their status; (2) humanitarian need of undocumented migrants, as well as of others including documented migrants, refugees and displaced people who are sometimes vulnerable and living in precarious situations; and (3) the need to ensure heath security globally and nationally, which in the case of a global pandemic requires operating on the basis that, for vaccination strategies to succeed in fighting a pandemic, the highest possible levels of vaccine uptake are required. Yet some population segments have had limited access to mainstream health systems, both prior to as well as during the COVID-19 pandemic. People with irregular resident status are among those who face extremely high barriers in accessing both preventative and curative health care. This is due to a range of factors that drive exclusion, both on the supply side (e.g., systemic and practical restrictions in service delivery) and the demand side (e.g., in uptake, including due to fears that personal data would be transmitted to immigration authorities). Moreover, undocumented people have often been at increased risk of infection due to their role as “essential workers”, including those experiencing higher exposure to the SARS-CoV-2 virus due to frontline occupations while lacking protective equipment. Often, they have also been largely left out of social protection measures granted by governments to their populations during successive lockdowns. This article reviews the factors that serve as supply-side and demand-side barriers to vaccination for undocumented migrants and considers what steps need to be taken to ensure that inclusive approaches operate in practice.

## 1. Introduction

Around the world, migrants and refugees often experience a wide range of formal and informal barriers to accessing health services [1,2]. This is especially the case for people with irregular residence status, or ‘undocumented’ migrants—i.e., non-nationals who enter or stay in a country without the appropriate documentation [3,4]. Barriers were often increased by constraints imposed in the course of efforts to contain the COVID-19 pandemic [5,6], within the broad variety of national adaptation policies [7] that were rapidly established as the emergency unfolded. These included restrictions on movement and assembly, as well as reductions in opportunity for employment, especially in the informal sector. Many services were transferred online, including a preference where possible for virtual consultations with health professionals. Undocumented migrants may lack the necessary hardware, software or connectivity to engage in such consultations [8], or not have the required health service registration or credentials to qualify for accessing the online booking systems.

Barriers to accessing health services are of particular concern in the case of undocumented migrants because past experiences and current situations often increase their susceptibility to disease and vulnerability to a range of health threats. These may arise from their living conditions, which may be overcrowded and lacking adequate sanitary facilities; from food insecurity, reliance on public transportation for mobility, absence of locally accessible medical services, communication barriers between physicians and im-migrant patients that limit access to health information and services (including key messages on COVID-19 prevention and health-seeking behaviour), and digital illiteracy and/or communication barriers that limit access to telemedicine [9,10,11,12]. Moreover, environmental factors, such as long-term average exposure to fine particulate matter having a diameter of less than 2.5 μm (PM2.5), may also influence COVID-19 outcomes [13]. Working conditions are a further important factor contributing to vulnerability. These may be poorly paid, poorly regulated and lacking adequate safety features, or involve exposure to physical, chemical or biological hazards. Migrants and refugees in some countries are concentrated in “essential” areas of work, such as in transport and in health and welfare services, as well as in hospitality and cleaning services, in all of which there are elevated risks of infection [14,15]. Owing to their precarious status, many continued to work to survive during the COVID-19 pandemic. However, they often struggled to follow measures to protect themselves from the virus, often with limited access to adequate in-formation, protective equipment or ability to physically distance. 

The temporal dimension has been of major significance both for the global COVID-19 situation and for the circumstances of undocumented migrants. Globally, reported new cases of COVID-19 infection [16]. rose in several waves from early 2020, reaching 750,000 cases/day by August 2021. This was followed by a further very steep rise after the appearance in late 2021 of a new virus variant, Omicron SARS-CoV-2, which was an order of magnitude more infective while causing a smaller proportion of severe infections requiring hospitalization and intensive care. As a result, there was a further peak of reported new COVID-19 cases at 3.8 million/day globally in January 2022 [16]. While these figures illustrate the relative speed of spread of the pandemic, they under-represent its true scale, since there has been substantial under-counting of infections and the accurate figures are likely to be much higher [17,18]. National responses varied greatly in both the speed of reaction and the extent of use of public health measures including requirements for screening mask wearing, social isolation, quarantining, lockdowns and border closures aiming to prevent the arrival and spread of infection. 

From the end of 2020, national health authorities in a number of countries were be-ginning to deploy the newly-developed COVID-19 vaccines, which had high levels of demonstrated efficacy for COVID-19, providing strong protection against serious illness, hospitalization and death. By January 2022, 11.5 billion doses of the nine WHO-approved vaccines had been administered globally, equivalent to almost 150 shots for every 100 people. However, the distribution of this use had been heavily skewed towards high-income countries [19].

While vaccination was offering a route out of the pandemic, barriers to success in following this route were also becoming apparent. As one of the largest public health emergencies of modern times, COVID-19 stress-tested many facets of health systems nationally and globally, exposing, among other areas, inequalities in access to and benefit from health services both within and between countries [20]. It has cast fresh light on the commitments made in human rights instruments affirming the right to health of all people, irrespective of their status, as well as in the United Nations (UN) Sustainable Development Goals (SDGs) to leave no-one behind. Experience with the roll-out of COVID-19 vaccines has underscored known, fundamental features of vaccination in public health: at the population level, an effective vaccination campaign requires a sufficiently high level of uptake and, at the individual level, everyone possible should be vaccinated for their own protection. Experience has also demonstrated the gulf between rhetoric about solidarity and collaboration on one hand and the reality of vaccine hoarding and the prioritization of national interest [21] on the other. 

Within the rapidly changing landscape [22], data on COVID-19 infection rates among migrants in general has been limited and the situation of undocumented migrants has received little attention. Studies, largely conducted in high-income countries [23,24,25,26], have shown that migrants, including more precarious migrants, may have numerous risk factors and vulnerabilities for COVID-19 and may have had high rates of infection and adverse clinical outcomes [27,28,29,30] and to be associated with higher rates of COVID-19-related mortality. This has been attributed to a combination of their living and working condition, often employment in high-risk, low-paid health and care settings, restricted access to healthcare, as well as limited awareness of prevention measures, often in the absence of consistent COVID-19 health information strategies for culturally and linguistically diverse communities. They continued to have higher infection rates even when waves of infection were subsiding in the host country [24,27].

Excluding migrants undermines the success of COVID-19 vaccine rollouts [31,32]. By September 2021, refugees and internally-displaced persons in high-risk categories had started receiving their first vaccination shots in 121 countries—but actual numbers vaccinated were very low in many hosting countries, with vaccine scarcity and vaccine hesitancy being contributing factors, according to the UN High Commissioner for Refugees (UNHCR) [33]. Displaced people are often last in line for vaccination [34]. Low- and middle-income countries host 85% of refugees and the first step in achieving equitable access to vaccines for them must be ensuring that these countries have enough doses of vaccine for them, as well as for the communities who host them [35]. It is also critical to invest in systems for vaccine delivery, including equitable roll-out in the last mile.

Access to COVID-19 vaccines is an equity issue [36,37,38,39] that extends beyond the categorization of being ‘undocumented’ or a migrant, and also has intersectional dimensions, including discrimination. For instance, gender plays a significant role [40], with women in humanitarian and displacement contexts being vaccinated at lower rates than their male counterparts. Historical experiences are among factors influencing COVID-19 vaccine uptake among minority ethnic groups [41,42]. Those held in detention centres may be denied any access to the vaccines [43]. The multiplicity of factors that may operate in combination emphasizes the need for using an intersectionality framework in addressing inequalities of access to COVID-19 vaccination by undocumented migrants [44].

A complex interplay of factors operates at the interface between access to vaccination on the one hand and uptake on the other. On the supply side, policies towards provision and its relative prioritization for undocumented migrants, especially when supplies are very limited, are often determined in the context of attitudes towards migrants and refugees and the extent to which there is a hostile or inclusive environment. The phenomenon of ‘crimmigration’ has been noted [45], in which a progressive amalgamation is seen between penal law practices and civil and administrative law processes, blurring the boundaries and undermining trust in authorities. The COVID-19 pandemic has exacerbated this, with health security being used as an additional factor in criminalizing immigration and intensifying migration controls. At a logistic level, willingness to offer vaccinations to undocumented migrants without risk of detention or deportation may be compromised by administrative blind spots [46] and red tape [47]. Fear of detention and expulsion, lack of trust in authorities [48], confusion about regulations, entitlements and protections, as well as difficulties in negotiating registration systems for vaccines, all add to levels of vaccine hesitancy [49,50] among migrants in irregular situations.

Within this complex nexus, the degree to which undocumented migrants have been vaccinated against the SARS-CoV-2 virus causing COVID-19 can serve as an indicator of several factors. These include (1) whether international and national commitments to human rights, including the right to health, and to humanitarian action are being observed; (2) the extent of influence of social and professional attitudes on service access; and (3) the extent to which personal knowledge, expectations and apprehensions on the part of undocumented migrants influence their uptake of COVID-19 vaccination where this is available [8,50,51,52,53,54]. As an example [50], in a study in four locations (Geneva, Baltimore, Milan and Paris) in February–May 2021, 14.1% of participating undocumented migrants reported prior COVID-19 infection and 26.2% reported fear of developing severe COVID-19 infection. Self-perceived accessibility of COVID-19 vaccination was high (86.4%), yet demand was low (41.1%) correlating with age, comorbidity and views on vaccination. It was concluded that public health interventions using different communication modes should build on trust regarding vaccination in general to tackle undocumented migrants’ hesitancy towards COVID-19 vaccination with a specific attention to men, younger migrants and those at low clinical risk of severe infection.

Understanding these factors can also assist in pointing to ways to overcome barriers and developing pathways towards future equity in the provision of health services, including vaccination, for undocumented migrants and other groups in situations of vulnerability, as well as better pandemic preparedness and enhanced global health security for all.

In the present review, barriers to vaccination that are relevant to undocumented migrants are examined both in relation to the availability of these services for the general population in different localities and in comparison with the services available to the wider migration groups, including refugees, asylum seekers, and displaced persons, of which they are a subset. The geographical perspective serves to highlight a crucial feature of the issue. Once vaccines were developed, large variations in access were seen between countries, which could be related to a set of factors including where vaccines were developed and manufactured and which countries had the resources to procure them and the delivery systems to roll them out. Those countries that invested in vaccine development and made advanced purchase commitments were at the head of the queue. As manufacturing became well established, there was pressure both for support for vaccine procurement for lower-income countries and for the waiver of patent protections on vaccines, to allow more countries to manufacture their own supplies. The COVID-19 Vaccines Global Access facility (COVAX) was positioned to play a key role in this, but political support has not translated into the resources necessary and concerns about safety and quality have been raised [55,56,57,58,59]. The degree of equity in the distribution of COVID-19 vaccines between countries, as well as factors influencing both supply and demand within countries, form an essential part of the background to the vaccination of undocumented migrants.

The aim of the present paper is not to provide an exhaustive review, but rather to give an overview that identifies characteristic challenges regarding supply-side and demand-side barriers to vaccination of undocumented migrants. It highlights differences in the vaccination of undocumented migrants that exist between countries and gaps within countries between their laws and policies on one hand and outcomes on the other. The commonalities revealed in the nature of issues affecting vaccination of undocumented migrants—including those relating to rights, equity, laws, policies and practices—and the varying outcomes of initiatives aiming to address the issues, indicate the basis for a pathway to move forward. Thus, the significance of the paper is in drawing together from these experiences a set of practical actions that have proved effective in particular settings to address both supply-side and demand-side challenges in seeking to increase COVID-19 vaccination coverage for undocumented migrants.

## 2. Methods and Limitations

The present article draws on evidence from several sources. These include a World Health Summit M8 Alliance Expert Meeting in 2021 [60], which reviewed situation summaries related to the vaccination of undocumented migrants in several European countries, South Africa and the USA, explored the nature of barriers encountered and discussed potential strategies for improving future access. The situation summaries were extended and updated by literature searches on undocumented migrants and their access to and uptake of health services in general and COVID-19 vaccination in particular. Publications in peer-reviewed journals were prioritized, along with official reports presented by leading organizations working in the field, including international organizations such as WHO, International Organization for Migration, OECD, and the UN High Commissioner for Refugees.

However, this has been a very rapid-moving field and one in which the resourcing, conduct and reporting of academic studies in a period of highly restricted travel has inevitably lagged behind the changing situation. Moreover, undocumented migrants are often difficult to reach and may be reluctant to provide information. Therefore, account was also taken of unofficial and/or unverified situation reports from relevant non-governmental organizations (NGOs), working on the ground. These include OXFAM, Médecins du Monde and the Platform for International Cooperation on Undocumented Migrants (PICUM) which promotes social justice and respect for the human rights of undocumented migrants within Europe. NGO reports can serve to highlight specific cases and examples of vaccination activities and point towards areas for further study or action.

As well as the general paucity of data on COVID-19 vaccination of undocumented migrants, another limitation in the present study was the lack of consistent data between countries regarding undocumented migrants, with respect to matters such as infection rates, outcomes of infection, and vaccination rates. There was also very limited information in many countries on specific initiatives for the vaccination of undocumented migrants.

## 3. COVID-19 Vaccination and Undocumented Migrants: Some Country and Regional Experiences

From the first licensing of a Coronavirus vaccine in a Western country in December 2020, there was intense competition for available supplies at the international level leading to ‘vaccine nationalism’ [61,62] and often strict regulation of the timing of provision of vaccine doses at national and local levels, generally with emphasis on priority for vulnerable groups at risk of severe disease (e.g., persons who are older, with certain preconditions, chronically ill or immunocompromised) and frontline health personnel and other key workers. Globally, WHO’s Strategic Advisory Group of Experts suggested a tiered system to prioritize who should receive the vaccines, and identified low-income migrant workers, irregular migrants and those unable to physically distance themselves, including those living in camps and camp-like settings, as being among priority groups for the allocation of COVID-19 vaccination [63]. International organizations conducted advocacy work with governments to raise awareness about the obligations of States to include migrants in official vaccination plans [22,64]. At the European level, the European Centre for Disease Prevention and Control (ECDC) issued a report in December 2020 on COVID-19 vaccination and prioritization strategies, which included recognition of the vulnerability of groups with little ability to ensure physical distancing, including those living in migrant centres [65]. A later ECDC report [66] recommended strategies to increase vaccination uptake among migrant populations, including culturally and linguistically tailored and targeted public health messaging, co-designed with affected communities, translated into key migrant languages and effectively disseminated. UNHCR noted [22] that, by April 2021, 153 States had adopted vaccination strategies that include refugees, but that, in many parts of the world, actual immunization remained a challenge, largely due to the unequal availability of vaccines and the capacity of health systems to deliver them.

Given that reaching some migrant populations for routine vaccinations was problematic even before the arrival of the COVID-19 pandemic [67,68], how did undocumented migrants actually fare in those countries and regions where vaccines were becoming available? A scan of available data from European countries, the USA and South Africa revealed an extremely fragmented and incomplete picture, with diversity in policies and strategies as well as in practices, making the logistics of obtaining vaccination very challenging for many people. This section provides a sampling of the spectrum of local situations in these countries. A further scan of reports from other parts of the world, including Asia, the Caribbean and Latin America as well as OECD countries, showed very similar situations in these regions too [69,70,71].

### 3.1. Europe

Outside of emergency situations and other limited cases, a variety of types of obstacles to accessing primary health care and other services are encountered by undocumented migrants in most countries of the EU. They include formal limitations on entitlements to services and, even in those countries which have schemes in place for undocumented residents, there are often challenges in practice (e.g., administrative barriers, complex procedures, unclear rules). Pervasive fear of authorities is often present, because of their history of prioritizing immigration status in granting entitlements and the direct immigration enforcement consequences of accessing mainstream health care. The risk of immigration enforcement relates to the broader policy environment, which in many countries criminalizes irregular status with the aim of expulsion and may also criminalize those who assist undocumented people, in order to deter irregular migration. The exclusion and mistrust created have important consequences for the COVID-19 vaccine rollout, both for the provision of and the choice to access vaccination.

As vaccines began to be rolled out, international organizations including the International Organisation for Migration (IOM) and European Union (EU) bodies including the ECDC and European Commission (EC) recommended addressing marginalized communities, including migrants in situations of vulnerability, in national vaccination strategies. However, national approaches in Europe varied substantially with respect to undocumented migrants [72]. Moreover, even where vaccination was officially available, if undocumented status was criminalized, many would avoid contact with government bodies, despite feeling unwell or being at risk of severe illness [73,74].

A study by the Pew Research Center [75] published in 2019 estimated that 3.9–4.8 million migrants (including asylum seekers awaiting decision on their status) were living undocumented across Europe in 2017. Excluding asylum seekers, the estimate was 2.9–3.8 million undocumented migrants. By either estimate, half the undocumented migrants were in Germany and the UK and over 70% were in those two countries as well as France and Italy. Expanding the list to include the Czech Republic, Greece, Spain and Switzerland covered 90% of the undocumented migrants in Europe.

The following sampling profiles the situation regarding COVID-19 vaccination of undocumented migrants in the four countries where they are estimated to be present in the largest numbers (France, Germany, Italy, UK) and also highlights some examples of important policies and practices from other European countries, which also host undocumented migrants, that indicate potential for successful strategies to extend vaccination to these people.

#### 3.1.1. France

The French health ministry affirmed [76] in January 2021 that vaccines would be available, free for all, regardless of residence status and with no requirement for a health insurance card, to all people living in France [77]. COVID-19 vaccination followed a 5-stage plan, with early stages prioritizing individuals based on their age and extant, high-risk comorbidities. The recommendations for prioritization of vaccinations [78] did not refer to undocumented migrants, but identified people in precarious situations as a target group to obtain the vaccine, although without according it a high priority, as these populations were not scheduled until Stage 4 in the vaccination national plan. Vaccination was non-mandatory and free for everyone and did not require residence papers, identity documents or public health insurance. It could be booked online or by phone, but PICUM’s situation report on France in October 2021 [79] referred to cases where some operators were unaware of the regulations and refused to book an appointment because a health insurance number could not be supplied. Some local health authorities allocated funding to the vaccination centres to operate mobile medical teams to visit locales where people in precarious situations live. A number of NGOs also operated mobile teams, either to provide vaccinations or for information and awareness raising [80].

The French National Public Health Agency developed an evidence-based vaccination strategy for individuals living in precarious conditions by a knowledge mobilization process for implementation in 2021. This approach facilitated the co-development of a COVID-19 information tool accessible to social workers and health mediators and provided a collective overview of the interdependent social and health issues, notably via a collective consultation process, to better understand difficulties in vaccinating populations without social security numbers [80].

While data protection is strong in France and officially there should not be a risk of immigration consequences, in practice fear of this has deterred many undocumented people from getting vaccinated [79]. Other practical barriers that undocumented people have faced in getting vaccinated in France include language (many vaccination points lack interpreters), and lack of adaptation of the vaccination campaign to specific living and working conditions of many undocumented people (e.g., working informally, with long or unsocial hours which are incompatible with the opening hours of the vaccination points). Additional barriers were lack of awareness of the specific basic needs (food and water supply), as well as low health literacy, low perception of the threat of COVID and low perception of the usefulness of vaccination, or circulation of fake news about the vaccines themselves. Access to vaccination has been especially problematic for unaccompanied children, with contributing factors being a variety of administrative barriers (e.g., parental authorization) and use of age assessment procedures that may be biased and inaccurate. Requirement for possession of a certificate of COVID-19 vaccination or negative COVID-19 rapid test in the past 24 h (“health pass” [81] for entry to many places and services (including hospitals) also created challenges for undocumented migrants, in the light of barriers to the vaccines [79]. For instance, while an electronic or paper certificate could be obtained at the time of vaccination, if lost it could be very difficult to obtain a replacement due to administrative hurdles linked to the ad-hoc health number. The complexity of converting ‘out-of-country’ COVID-19 vaccinations to French vaccination certificates has also created administrative barriers for undocumented migrants.

#### 3.1.2. Germany

Médecins du Monde in Germany noted that, at an early stage in the pandemic, many of their clients had limited access to information (mainly due to language barriers), testing, prevention (e.g., masks, disinfectant, appropriate housing conditions) and treatment [82]. Undocumented migrants in Germany have legal entitlement to restricted health services covering acute illness and pain and other emergency care. However, public authorities, including social service departments which need to be approached for covering the cost of these health services, have a duty to report undocumented persons, who therefore avoid seeking medical care.

Germany’s COVID-19 vaccination regulation [82] entitled everybody with habitual residence or registered address in Germany to get vaccinated, prioritizing asylum seekers in the second priority group (after people above the age of 80 and healthcare workers). Undocumented migrants were not explicitly mentioned in the regulation. Subsequently, however, the Ministry of Health confirmed after a parliamentary request that undocumented migrants living in Germany are entitled to vaccination. The duty to report is no major barrier to vaccination in principle, as doctors are excluded from the duty to report and no application for cost coverage is necessary [74,82,83,84]. In practice, however, access to vaccination is very inhomogeneous: often, vaccination as a priority group was only possible with a personal invitation letter, which of course depends on a registered address. Many vaccination centres required a personal ID. Concerns and practical issues remained, most notably around providing proof of habitual residence and around the fear that data could be shared with immigration enforcement authorities.

Against this background, the degree of vaccine hesitancy emerges as a complex picture. A cross-sectional study of Turkish- and German-speaking citizens in Munich reported in 2021 that COVID-19 vaccine hesitancy was much higher among people with migratory backgrounds [50]. However, a wider study reported in February 2022 [85,86,87] observed that, while overall vaccine uptake among people with a migration history was a little lower (84%) than among the rest of Germany’s population (92%) at the time, their willingness to obtain a first dose might actually be higher. Factors contributing to vaccine uptake included higher socioeconomic status, higher age, less experience of discrimination in the health care and nursing sector, good knowledge of German, trust in the safety of the vaccination and the health system, and viewing vaccination as a communal measure. Factors contributing to vaccine hesitancy included prior experiences of discrimination and language barriers, as well as false information about COVID-19 and vaccines, which tend to have a higher circulation among immigrant communities than non-immigrant ones. Significant differences in levels of vaccination were observed between different cities in Germany. Bremen’s high rate of vaccine uptake among immigrants was associated with provision of mobile vaccination teams, deployment of health professionals at food banks who speak the immigrants’ mother tongues and distribute information material in 12 different languages, and pragmatism in providing vaccinations despite lack of proof of identity.

#### 3.1.3. Italy

The Italian Immigration Act explicitly guarantees access to urgent or essential health care, including vaccination as part of preventive public health care campaigns, to all people living in Italy, including irregular migrants. However, non-Italian citizens received comparatively late diagnosis of COVID-19 and were more likely to be hospitalized and admitted to intensive care, with an increased risk of death in those coming from lower human development index countries [88]. While Italy’s COVID-19 vaccination strategy [89] does not mention undocumented migrants explicitly, the Italian Medicines Agency guidelines [90] make clear that undocumented people are entitled to COVID-19 vaccination, in line with inclusion [22] and the right to health.

As the law forbids any health data sharing with police and judicial authorities, undocumented persons are theoretically able to access COVID-19 vaccination without fear of immigration consequences. However, in mid-2021 the online booking platforms, which are managed by Italy’s 20 regions, in most cases still required information and documents which were unavailable to most migrants with irregular residence status. Wide variations were seen between regions, with some (e.g., Apulia, Campania, Sicily, Veneto, Lombardy) enabling undocumented migrants to book their vaccinations online. There have also been local initiatives aiming to reach out to undocumented people to vaccinate them. For example, the humanitarian organization Emergency organized COVID-19 vaccination campaigns in the towns of Ragusa and Vittoria for all, especially for undocumented migrant farmworkers [91]. Since the vaccination centres tend to be set up in towns and cities, there remains concern about reaching less accessible undocumented migrants—for example, those who find agricultural work and live precariously in informal dwellings. A study of access to COVID-19 vaccination during the pandemic in the informal settlements of Rome [92] observed in October 2021 that the percentage of vaccinated people there, ranging between 4.4% and 55.5%, was significantly below the vaccination rate of Italy’s population (close to 80%), with particular attention needing to be paid to transiting and irregular migrants who were at greater risk of lacking access to vaccination.

#### 3.1.4. United Kingdom

Despite the operation of a ‘hostile environment’ policy towards migrants for several years [93], the UK’s COVID-19 vaccine delivery plan, issued in January 2021, accorded migrants living in the UK eligibility to receive COVID-19 vaccines free, regardless of their legal or undocumented status. Undocumented migrants were not initially cited as a priority group [23], but by the end of the year were being listed as one of a number of at-risk groups being targeted for vaccination [94]. Government guidance stated that no immigration check would be required in the context of the vaccination [95]. The pent-up demand for vaccination was evidenced when thousands of undocumented migrants turned up at a pop-up clinic in the center of London in May 2021, having been encouraged and assured that no details or identity information would be passed on to the police or immigration [96]. A month later, one of the poorest boroughs in London opened a walk-in clinic for people without documents, temporary migrants, asylum seekers and homeless people [97].

However, historical policies towards migrants who do not have ‘leave to remain’ had generated distrust and insecurity among undocumented migrants and practical barriers remained [98], including from some General Practitioners who would not register undocumented migrants as patients [99]. This made it more difficult to obtain the NHS number needed to book vaccinations, which is mainly done online [100]. A national qualitative interview study completed in March 2021 explored the views of undocumented migrants, asylum seekers and refugees, using the ‘3 Cs’ model (confidence, complacency and convenience) to explore COVID-19 vaccine hesitancy, barriers and access [101]. It revealed concerns over vaccine content, side-effects, lack of accessible information in an appropriate language, lack of trust in the health system and low perceived need. Barriers to accessing COVID-19 vaccination were reported and concerns expressed about being excluded from or de-prioritized in the roll-out. Undocumented migrants described fears over being charged and facing immigration checks on presenting for a vaccine. Those interviewed after the government announced that COVID-19 vaccination could be accessed without facing immigration checks remained unaware of this. Convenience of access was a key factor in deciding whether to accept vaccination, along with accessible information on vaccination.

A survey carried out in May 2021 among UK refugees, asylum seekers, and undocumented migrants [102] found that many cited family and friends as the most common source of COVID-19 vaccine information, with discouraging information from a range of sources acting as significant deterrents to vaccination. Misinformation spread unchallenged as a result of lack of access to reliable information in appropriate languages and formats. As well as worries about the vaccine, there was also distrust in the systems and organizations responsible for its rollout and delivery. Survey participants identified a number of facilitators to increase COVID-19 vaccine confidence and uptake, These included routing reliable information via trusted organizations and individuals, opportunity for discussion with health professionals, access to information resources commensurate with language and literacy needs and in a variety of formats, targeted webinars and public information sessions, accessible pop-up vaccination clinics in the local community, informed peers and community and religious leaders acting as vaccination champions, involvement of local voluntary, community and social enterprise organizations, communications directly addressing circulating misinformation, and provision of reassurance that data collected for vaccination purposes would not be shared for immigration enforcement.

#### 3.1.5. Other European Countries

After France, Germany, Italy and the UK, the four European countries with the next highest numbers of undocumented migrants at the outset of the COVID-19 pandemic were the Czech Republic, Greece, Spain and Switzerland.

In the Czech vaccination strategy published at the end of 2020, only people with public health insurance were able to access COVID-19 vaccination. Persons without residence or with insecure legal status (such as refugees, asylum seekers, and undocumented migrants) were not recognized as a priority group) [103]. On 11 June 2021, access was extended to include regularly residing migrants with private health insurance. Undocumented migrants were effectively excluded from both provisions, since they would not be able to use the booking systems requiring data relating to identity, residence and insurance status. At the end of June 2021, the Ministry of Health launched vaccination registration for those foreigners residing in the country on a long-term basis who lacked access to the public health insurance system. This required an identity document, residency permit and payment of about €30. A letter from the Ministry of Health to regional authorities stated that they could also vaccinate migrants without health insurance. However, it did not explain how to reach this group, how to practically organize their vaccinations, or how the regions would pay for this. It was also unclear whether information would be transferred to the immigration authorities [104].

When Greece began deploying COVID-19 vaccines at the end of 2020, its vaccination strategy did not initially refer to undocumented migrants. Refugee and asylum seekers experienced higher rates of COVID-19 infection than the general population [105]. Asylum seekers in mainland camps were provided with vaccinations, but not those who received a second rejection of their asylum application. Up to September 2021, registration for vaccination required a social security number. There was no guarantee that people without documentation who sought vaccination would not be reported to the immigration authorities. However, a new law published in October 2021 included provision for undocumented and stateless persons in Greece to obtain a provisional social security number enabling registration for COVID-19 vaccination and receipt of vaccination certification and COVID-19 self-tests. This law also indicated that undocumented migrants would not be deported as a result of accessing the vaccination process. It foresaw roles for NGOs and municipalities in the delivery of COVID-19 vaccination, but did not lay out practical implementation steps [106,107]. Delays in implementation were being reported early in 2022 [108].

Undocumented migrants and other marginalized populations, including those in a vulnerable socioeconomic situation, are explicitly cited in the Spanish federal vaccination strategy as groups to be vaccinated, although they are not identified as a priority group and there is no detail on how to reach them. They are also mentioned in several regional strategies, which implement the federal framework and, in some cases, work with NGOs. Legally, access to health care is provided for undocumented people who can prove they have lived in Spain for at least 3 months, and that their country of origin doesn’t cover their medical expenses (which requires a certificate from the country of origin that may be difficult or impractical to obtain). Online booking systems, managed regionally, vary in their requirements but generally require a valid health insurance number which can be complex to obtain for undocumented migrants in different situations. Groups of concern who may be missed from vaccination include hard-to-reach undocumented people such as those working in agriculture, sex workers, homeless people and people in detention centres, where there are no unified COVID-19 protocols [23,109]. While Spain had reached a COVID-19 vaccination coverage approaching 90% by March 2022, there were continuing reports of bureaucratic obstacles and deportation fears delaying vaccines for the undocumented [110].

Undocumented migrants in Switzerland are, like all other residents, legally required to take out health insurance. They are also entitled to premium reductions and to access basic Swiss healthcare services. The health insurance companies, as well as health care professionals, are obliged to accept undocumented migrants and are not allowed to pass on information about them [111,112]. There is a paucity of information about the actual extent of COVID-19 vaccination of undocumented migrants in Switzerland, but as an example of provision, the Canton of Zurich has a center, Meditrina, as a medical contact point for undocumented migrants, where they can obtain access to COVID-19 vaccinations [113].

A situation report to the UN Special Rapporteur on Human Rights of Migrants by PICUM [114] in June 2021 observed that several EU countries (including Belgium, France, Ireland, Italy, the Netherlands, and Spain) had implicitly or explicitly included undocumented migrants in their vaccination strategies and taken practical steps to facilitate their access to the vaccines. However, other member states (such as the Czech Republic, Hungary and Poland) had explicitly excluded undocumented people from their vaccination strategies or imposed burdensome or impossible administrative requirements. The project ‘Vaccinating Europe’s Undocumented’, which also reported in 2021 [115], assessed ways in which undocumented people are included in or excluded from COVID-19 vaccination programs implemented by European countries since December 2020. Scorecards were constructed for 18 countries, covering policy transparency, undocumented access, identity and residency requirements, marginalized access and privacy guarantees. The two highest scorers were the UK and Portugal, which earned the label “Open and Accessible”. These were the only ones gaining positive scores in all categories, evidenced by finding written material confirming that vaccination plans and strategies are publicly available; undocumented people are given access to the vaccine; identification and residency requirements are specified; other marginalized groups are also granted some level of access to vaccination; and there is a good level of privacy guarantees for those who do get the vaccine. By contrast, Slovakia, Czech Republic and Poland, all of which are explicitly exclusionary, were labelled “Closed Door” countries. The countries in between (Austria, Belgium, Denmark, Estonia, France, Germany, Greece, Ireland, Italy, Malta, Netherlands, Romania, Spain) were “Confused”, offering insufficient public information for clarity in the five areas examined.

As an example from the middle group, Belgium’s weak overall assessment as “Confused” is consistent with a range of reports before and since the scorecard was issued. In early 2021, both the Brussels and Federal health ministers affirmed that it was out of the question to exclude undocumented people from the vaccination process [74]. Subsequently the Brussels-Capital region developed strategies to implement its vaccination campaign to include undocumented migrants and other groups facing social exclusion. ’Mobivax’ mobile teams were established, coordinated by a consortium of non-governmental organizations (NGOs) including Médecins du Monde, Médecins Sans Frontières, the Red Cross, and the Samusocial. Each team included two cultural mediators to facilitate dialogue with the target group. They began vaccinating homeless and undocumented people in the region on 19 May 2021, administering the Johnson & Johnson vaccine, which can be given in a single dose. To receive the vaccine, undocumented migrants in Belgium were required to possess a national number, available from GPs and local authorities without immigration consequences. However, in practice it was not always easy for the homeless to obtain the national number and some NGOs have called for clear and binding policy frameworks formalizing the arrangement and for improved communication and awareness-raising around this issue by NGOs and government [116]. Difficulties encountered by undocumented people in Belgium in getting vaccinated and obtaining proof of vaccination were still being reported in February 2022 [117].

Other European countries not included in the scorecard also show a distribution along this spectrum, with policies and practices likely to place them in the center category [23]. For example, the Finnish vaccination strategy does not specifically mention undocumented migrants but includes free vaccines for everyone aged 5 or over in Finland [118]. The Finnish Ministry of Social Affairs and Health recommended in March 2021 that municipalities grant access to free COVID-19 vaccines to undocumented migrants. PICUM has noted [119] that, in practice, undocumented migrants may face difficulties in accessing mainstream vaccination points, since booking online requires official identification, which is largely not available for an undocumented person in Finland, and booking by phone can be challenging if the patient doesn’t speak Finnish, Swedish or English and does not know the health care system. One of the two alternative avenues for undocumented migrants to access the vaccine are drop-in vaccination points, opened at a later stage of the national vaccination strategy. Access requires having a temporary health number, which can be obtained from health care providers at the first point of contact, or one on the spot at the vaccination center. In the second alternative, mobile teams composed of public health care professionals administer the vaccines at sites operated by NGOs on specific dates. In Helsinki, these are deployed at two specific sites where undocumented migrants are known to live. The migrants are given information and encouragement by social workers and NGOs to attend, with reassurance that health care professionals are bound by strict confidentiality rules and can be trusted to not share personal data for purposes not related to health care. Nevertheless, some undocumented migrants, especially beyond Helsinki, may be outside these social and NGO networks; there are continuing fears about contacts with public authorities and around the vaccination itself; and there are administrative barriers where there are no official guidelines for health care professionals about health care entitlements. Finland’s requirement for certificates of COVID-19 vaccination, especially in response to the spread of the Omicron variant, have also provided a challenge for undocumented migrants since they are unable to obtain an electronic certificate from a dedicated website. They can, however, receive a paper certificate on-site after their vaccination.

In general, undocumented people in Norway can only access emergency health care and “health care that is totally necessary and cannot be deferred”. Norway has seen higher than expected proportions of migrants among COVID-19 cases, amounting to twice the population’s share [28,120], highlighting vulnerabilities in this group. Undocumented migrants are not mentioned explicitly in the Norwegian vaccination strategy, but all people regardless of residence status are legally entitled to vaccinations. In addition, in spring 2021, the Norwegian Directorate of Health called on all local and regional health authorities to make vaccination available for everyone. In practice, however, booking an appointment for vaccination requires a valid personal identification number, and being registered with a GP. Both are broadly unavailable to undocumented migrants. In large part, the chance of undocumented migrants being vaccinated is determined by the local municipalities and their motivation to find ways around bureaucratic barriers. The risk of immigration consequences when accessing the vaccines is virtually non-existent since it is illegal for medical staff to report undocumented patients [121].

At the more exclusionary end of the scale, the Hungarian vaccination strategy does not mention undocumented migrants. The vaccination booking system requires both a valid social security number, which is not available for undocumented people, and a registered home address, which can be very difficult for them to prove. Furthermore, civil society has warned of indications that the registration data will be checked against data held by the immigration authorities [122].

As a European country that is not a member of either the EU or the European Free Trade area, Turkey has often been neglected in surveys and reports concerning COVID-19 vaccine coverage on the continent. Nevertheless, Turkey merits close attention. In the last several years, Turkey has experienced a major flux of migrants and refugees, some passing through on their way to Europe and others remaining. Turkey hosts more international refugees and displaced persons than any other country—in April 2020 amounting to around 3.7 million refugees from Syria, 370,000 asylum seekers and refugees under international protection in Turkey, most of whom were from Afghanistan and Iraq, and around 455,000 irregular migrants mainly from Afghanistan, Pakistan and Syria [2]. The Turkish Government evolved a long-term strategy of ‘harmonization’ of the refugee population, underscoring a two-way process of adaptation and mutual learning on the parts of both host and refugee communities, with cities playing and important role. A mid-2020 report highlighted the important role that Turkish civil society organizations (CSOs) had been playing in helping meet the humanitarian needs of refugees and how this was affected by the COVID-19 pandemic and the strict measures introduced by the Government to contain it, with many CSO activities being suspended [123]. In April 2020, the Turkish Government announced that COVID-19 related health services, including access to personal protective equipment, diagnostic testing and medical treatment, would be provided free regardless of registration status, facilitating access to health services during the pandemic for irregular migrants. A number of reports have highlighted difficulties faced by undocumented migrants in Turkey in accessing treatment for COVID-19 and, since they became available, vaccines. Examples [124,125,126,127] include language barriers leading to difficulties in accessing health information (countered by distribution of health information leaflets in Arabic); vaccine hesitancy; exclusion of migrants without ID cards from hospital treatment; and reluctance of unregistered migrants to visit health institutions because of fears of deportation or loss of accommodation or employment, with reporting of undocumented migrants by hospital staff remaining mandatory. A survey of refugees aged 65 and above in Turkey, conducted in February 2021, noted that at this early stage in vaccine roll-out, although they were in the target group for vaccination, only 11% of the interviewees had been vaccinated [128].

### 3.2. South Africa

In general, African countries experienced four waves of COVID-19 infection, typically peaking around July-August 2020, January and July–August 2021 and December 2021–January 2022 [16,129]. At the beginning of March 2022, 11.55 million cases had been reported, the largest component being South Africa’s 3.68 million [16,130]. South Africa had reported a total of 99,500 COVID-19 related deaths at that time—a significantly lower death rate than seen in, for example, France or the UK [16].

Lower reported COVID-19 case numbers and deaths in Africa, compared with other regions (only Oceania has a lower death rate), have been linked with a complex variety of factors. These potentially include poor reporting systems, limited testing capacities, public health infrastructures and mitigation strategies, weather conditions, lower volumes of air travel, young populations (median age in Africa is 19.4 years, compared with 40 in Europe and 38 in USA), previous exposure to other locally circulating coronaviruses, malaria co-infection, genetic factors for severe COVID-19 illness, strong political will and swift imposition of lockdowns [131,132,133]. The picture in many African countries when COVID-19 vaccines became available is typified by the example of Uganda [134]. Its government policy made COVID-19 vaccines available to all, including the more than 1.4 million refugees living in the country, but the actual vaccination rate among refugees was very low. As well as vaccine hesitancy, many logistic problems were encountered by refugees, including vaccines not reaching the places where they live and work, administrative hurdles, and insufficient information and outreach targeted to them.

South Africa, a middle-income country with the most industrialized economy in the region, has drawn migrants from many other countries. There were an estimated 2.9 million migrants residing there in 2020 (approaching 5% of the overall population)—the largest number of immigrants on the African continent [135]. Pull factors include opportunities for work in, among other areas, mining, manufacturing and agriculture, and push factors including poverty, adverse climatic changes and weather events and political and physical insecurity in other places.

Numbers of ‘undocumented’ persons in South Africa are thought to run to millions [136]. Estimates have been difficult to make as migration has long been a politicized issue complicated by xenophobia and waves of violence against immigrants [137,138], with language often framed in the context of ‘illegal’ persons and health security concerns to justify exclusion. In practice, the category ‘undocumented’ includes both ‘unauthorized migrants’ who have entered South Africa without official permits, and those who have entered with authorization but have encountered bureaucratic difficulties and delays in renewing papers [139,140].

Legally, migrants have a constitutional right to health in South Africa. Both legal and undocumented residents have the right to primary care, with the same entitlements being applied to refugees and asylum seekers as for all South African citizens. However, the health system is weak in South Africa and struggles to cope with the demand from the population as a whole. In practice, health care professionals in the country have routinely denied health care and treatment to many asylum seekers, refugees and migrants, with foreign-born residents encountering systematic discrimination in obtaining basic care and access to a range of public health services [141,142,143], including treatment for HIV and tuberculosis infections whose incidences are high in the southern African region.

South Africa’s response to COVID-19 reflected this history, with initial reticence to include asylum seekers, refugees and foreign migrants in policies and mechanisms to deal with the pandemic [140,144,145], followed by confusion over the actual policy [146] and evidence of xenophobia and discrimination, including gender-based, in treatment [146,147]. By September 2021, South Africa had administered more than 16 million doses of COVID-19 vaccines to its citizens. The Government was being urged from many quarters to prioritize refugees, asylum seekers and undocumented persons in the vaccination programme [148]. These groups, and especially the undocumented, had been effectively excluded by the registration requirements of the Electronic Vaccination Data System to provide an ID number, passport number or refugee/asylum seeker permit number [149]. The Government responded to these calls, and to the escalating infection rate as the Omicron variant took hold, by announcing in December 2021 that the vaccine was being offered to those without documentation [150], including through pop-up centres [151].

### 3.3. United States of America

The USA is the only country in the developed world without a system of universal healthcare. Its health system is fragmented both by source of provision (partly through health insurance provided either through the workplace or privately purchased, partly through Government schemes such as Medicaid, Medicare and Children’s Health Insurance Program) and geographically, with Federal approaches to healthcare services interpreted at the State level and often politicized [152]. According to a 2020 study by the Pew Research Center [153], there were 10.5 million unauthorized immigrants in the USA in 2017, which accounts for an estimated 3% of the total US population. About half of the estimated number of undocumented immigrants are uninsured, compared to 31 million US citizens in 2020 who are uninsured. Undocumented immigrants are systematically excluded from enrolling in federal health insurance programs mentioned previously and coverage through the Affordable Care Act (ACA) Marketplaces [154]. Additionally, evidence has shown that many undocumented migrants are less likely to have employer-based health insurance because they are often employed in industries that do not offer health benefits (i.e., agriculture and construction). The Personal Responsibility and Work Opportunity Reconciliation Act (PRWORA) of 1996 significantly transformed the US welfare system, where welfare was no longer seen as an entitlement to individuals [155]. Title IV of PROWRA specifically makes it difficult for immigrants to access welfare, stating that “self-sufficiency has been a basic principle of United States Immigration law since this country’s earliest immigration statues” [156]. The document further reads that the eligibility rules are to assure “that individual aliens do not burden the public benefits system”. During the COVID-19 pandemic, lack of health insurance left many undocumented immigrants unable or unwilling to seek treatment and the COVID vaccine.

The inability and unwillingness to access COVID-19 healthcare treatment and vaccine were exacerbated by immigration polices implemented after the 2016 elections [157]. For example, the public charge rule penalizes the use of sought public benefits by immigrants looking to change their immigration status. Since 2021, many of these anti-immigration policy changes have been revised, including the public charge rule, but these past immigration policies continue to have a “chilling effect” on healthcare coverage of undocumented migrants, especially during the COVID-19 pandemic [158]. Despite the new Administration supporting the expansion of health coverage, no specific proposal has been offered yet. Many undocumented immigrants with COVID-19 can seek treatment in emergency department services covered by Emergency Medicaid. The Emergency Medical Treatment and Labor Act (EMTALA) requires Medicare-participating hospitals with emergency departments to provide emergency care to all, regardless of immigration status, and in a non-discriminatory manner. However, EMTALA no longer applies when a patient is stabilized or when an uninsured patient does not require emergency services. Several states amended their Emergency Medicaid-qualifying conditions to include outpatient prevention, testing, and treatment of COVID-19 [159].

When the first approvals were given for COVID-19 vaccines in the USA, the US Centers for Disease Control and Prevention guidelines for COVID-19 vaccination accorded high priority to, among others, health-care personnel and other essential (both health and non-health workers [160]), groups in which nondocumented migrants are prominent. A Department of Homeland Security statement [161] in February 2021 said that the Government “fully support equal access to COVID-19 vaccines and vaccine distribution sites for undocumented immigrants. It is a moral and public health imperative to ensure that all individuals residing in the United States have access to the vaccine. DHS encourages all individuals, regardless of immigration status, to receive the COVID-19 vaccine once eligible under local distribution guidelines”. The statement announced that, as well as fixed clinics, there would be pop-up or temporary vaccination sites, and mobile vaccination clinics. It assured that Immigration and Customs Enforcement and Customs and Border Protection would not conduct enforcement operations at or near vaccine distribution sites or clinics. However, a Congressional Research Service, report later in February 2021 observed [162] that barriers to vaccination remained among the unauthorized population, including a range of sociodemographic factors and prevalence of immigration enforcement fears associated with the risk of incurring a ‘public charge’. Socioeconomic and sociodemographic factors were also highlighted in the University of Minnesota collection on health, vaccines, and equity [163].

Recent evidence points to uptake of the vaccine being lower among immigrants and especially those who are undocumented [54]. In addition to fears over their precarious state, reasons given for vaccine hesitancy included the lack of access to information, language barriers and conflicts between work and clinic hours. Strategies to overcome these barriers included the use of trusted leaders and improved informational messaging from government agencies and the medical community, broad engagement of the community and responsiveness to language and cultural needs, as well as overcoming practical obstacles to vaccination access. A strong role was suggested for existing, trusted, culturally intelligent community-based organizations and local sociocultural processes.

## 4. Conclusions

Since COVID-19 vaccines became available at the end of 2020, evidence from many parts of the world demonstrates comparatively low vaccination rates for undocumented migrants, due to the operation of a combination of supply-side and demand-side barriers. On the supply-side, an essential precursor step to achieving equitable access to vaccines for all, globally, must be ensuring that low- and middle-income countries have enough doses of vaccines suitable for their use, including for refugees and displaced people, both documented and undocumented, as well as for the wider community [34,164]. In this regard, an ethical framework is needed for global distribution of COVID-19 vaccines [165] and the weaknesses revealed in the COVAX facility during the pandemic require urgent attention [166]. It is also vital to invest in local vaccination systems to ensure access by all to COVID-19 vaccination.

While attention has been given to the issues of comparative efficacy and safety of different COVID-19 vaccines since a number of vaccines began to appear, the specific relevance to undocumented migrants has received very little discussion, including the relative benefits versus risks to them of vaccination. Evidence of wide-ranging damage to a variety of organs has accumulated [167], including examples of kidney disease resulting from COVID-19 infection and leading to the need for renal replacement therapy [168,169]. However, in the specific case of undocumented migrants, such outcomes of COVID-19 infection, as well as of adverse effects of vaccination, have not been reported and remain to be investigated. It has been suggested that the single-dose Janssen vaccine may be advantageous for those who may prefer to avoid repeat visits to a vaccination clinic, but WHO nevertheless recommends a second dose, which results in increased protection against symptomatic infection and against severe disease [170]. Decreasing efficacy of different vaccines towards new variants of the SARS-CoV-2 virus may require additional doses and/or the development of next-generation vaccines, which must also be considered in the context or protection of individuals in precarious situations, including undocumented migrants.

Arguments in favor of ensuring equity of access to COVID-19 vaccination for undocumented migrants, fall into three main categories: (1) the right to health of all people, irrespective of their status; (2) humanitarian need of people who are vulnerable and living in precarious situations, and (3)) the need to ensure heath security globally and nationally, which in the case of a global pandemic requires operating on the basis that “no-one is safe until everyone is safe”. However, the commitments to health rights, health equity and the SDG principle of “leaving no one behind” have failed to translate into equal access to vaccines, both at the global level between countries and, in many countries, at the national level between different societal groups. In practice, humanitarian action has often been left to NGOs and CSOs, accompanied by limited government interest, insufficient resources and fragmented and incomplete coverage of needy people who are among the hardest to reach. Furthermore, many countries have been slow to recognize and adopt the collective health security argument, faced with high demand and strong competition for insufficient vaccine doses and/or limited delivery capacities, as well as politicization of migrant issues and sometimes use of health security as a pretext for exclusionary policies and tighter border controls.

Along a spectrum of approaches to the COVID-19 vaccination of undocumented migrants, most countries fall in the middle ground, which has been characterized by (often slow) official inclusion of undocumented migrants in policies for access, but generally at lower priority than other vulnerable or precarious groups. Implementation of access under the policy has often been very deficient, with administrative hurdles including registration systems requiring documentation that is absent/unattainable by the migrants or involving sites and schedules that are impractical for their living or working conditions. These barriers are sometimes compounded by hostile attitudes by administrators or health workers, whether due to underlying attitudes to migrants or ignorance of the legal position.

On the demand side, evidence from formal studies and informal reports from those working on the ground with undocumented migrants reveals a range of disincentives to accessing vaccination where it is offered. These include, as well as vaccine hesitancy due to concerns about the vaccine itself, fears that there are no clear-cut firewalls preventing information about them being passed to immigration authorities and which might result in imprisonment or deportation. Other barriers for the undocumented migrants include low awareness of entitlements and relevant access points, lack of information in appropriate languages, circulation of misinformation, fears about costs and about possible side effects, and practical difficulties in attending vaccination sites at available times.

Concrete solutions to the challenge of vaccination of undocumented migrants must therefore address both supply- and demand-side barriers, tailored to the specific conditions of the country and to the circumstances of the undocumented migrants themselves. Commonalities in the findings from many countries, summarised and cited in this paper e.g., [5,23,34,53,68,71,101,171,172,173,174] indicate a package of practical actions, based on experience, that can help to increase COVID-19 vaccination coverage for undocumented migrants:

### 4.1. Making the Case

Persuading policy-makers, health service providers, media and the public is imperative, with arguments based on a combination of the right to health, humanitarian need, the protection of essential workers, services and the functioning of the overall economy, and the imperative of achieving the highest possible levels of vaccination coverage to prevent resurgence and eliminate opportunities for development of virus variants.

### 4.2. Ensuring Consistency of Laws and International Commitments

Ensuring consistency of laws and international commitments: highlighting the international instruments on human rights affirming the right to health of all people, irrespective of their status, as well conventions on migrants and refugees and the UN SDGs, to which countries are signatories, and the need for alignment of national laws with these obligations.

### 4.3. Setting the Policies

Ensuring that policies on vaccination explicitly establish the rights of all in the country to be provided with COVID-19 vaccinations, irrespective of their residence or documentation status and with guarantees that personal details will not be passed to immigration authorities; and requiring consistency in the uptake and implementation of the national policies across all states, provinces and other sub-jurisdictions.

### 4.4. Ensuring the Vaccination System Is Genuinely Open and User-Friendly for Undocumented Migrants throughout the Country

Ensuring that the booking system has the least possible complexity, collects the minimum possible client information and the fewest essential documents or registration numbers, while providing assurances of a strong firewall that will prevent information about clients being passed to immigration authorities; that vaccination centres are at sites and have operating hours that are compatible with the locations and hours of availability of the patients, and that health workers are fully instructed in the rights of undocumented migrants; that information about COVID-19 vaccination is made available in languages and cultural forms that are compatible with the needs of undocumented migrants; that certification of vaccination status is made readily accessible to undocumented migrants in formats that they can readily access and use and that do not flag their undocumented status; and, through all these areas of action, engaging with undocumented migrants and their representatives to ensure that they are fully consulted about their needs and about the processes being designed to respond to them.

### 4.5. Addressing Vaccine Hesitancy

Forming alliances and collaborations between governments, NGOs, CSOs and community leaders and vaccine champions is essential to counter misinformation, promote understanding of the safety, need for and value of COVID-19 vaccination and reinforce the reliability of information firewalls that prevent sharing of information about the identity of undocumented migrants.

It is clear that the virus SARS-CoV-2 will be circulating for a long time to come. Dealing with this virus and its emerging new variants, as well as preparing for the next inevitable outbreak of a new viral infection in the world, needs to be a global priority in building resilience and limiting the costs to lives and economies. No strategy can succeed which is not inclusive of all people, including undocumented migrants.

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
