# Peer review of "The Challenge of Reaching Undocumented Migrants with COVID-19 Vaccination"

_ijerph, 2022, doi:10.3390/ijerph19169973_

Round 1
Reviewer 1 Report
I read with great interest the article entitled “The Challenge of Reaching Undocumented Migrants with COVID-19 Vaccination”
The work reviews the factors that serve as barriers to vaccination for undocumented migrants and considers what steps need to be taken to ensure that inclusive approaches operate in practice
Major concerns.
Although the subject of the study is interesting and well write, some aspects should be clarified.
1) A section concerning methods is totally absent. This shortcoming takes away the value of the article and brings it closer to a commentary than to a review. It is imperative that the authors write this chapter.
2) Any personal opinion based on political convictions must be removed from the text (i.e. Biden, Trump). The form must be that of a scientific journal not an information journal.
3) In the introduction what is at the basis of the analyzed problem is omitted: there are countries that still do not have access or have limited access to vaccines. [cfr Nioi, M., & Napoli, P. E. (2021). The Waiver of Patent Protections for COVID-19 Vaccines during the ongoing Pandemic and the Conspiracy Theories: Lights and Shadows of an Issue on the Ground. Frontiers in Medicine, 8. ] I believe it is necessary to dedicate a paragraph of the introduction to the theme.
4) The discussion is non-existent. The authors' arguments should be studied more clearly and linearly (need to allow easy access to vaccines for undocumented migrants). Even the opinions of those who believe that this is not a priority (and the relative refutation) must be reported in a linear fashion. The authors highlighted the issues. It seems necessary that in the paper prospect of concrete solutions that are not mere wishes.
5) We remind the authors that the current vaccines were prepared on a virus circulating in 2019. According to the authors, does it make sense to administer this vaccine considering that the kidney is not totally immune to the variants that have recently emerged?
6) The authors should report - at least in the answer - what is the importance of their work in terms of novelty and orientation of health policies. The mere description of Public Health situations in different States without solutions being proposed cannot be considered in a scientific journal.
Reviewer 2 Report
The paper reviews factors related to barriers and the situation regarding COVID-19 vaccination of undocumented migrants in the European countries (e.g..; France, U.K, Italy), South Africa and U.S. It also makes the case for policy-makers, health service providers, media and the public to develop policies as well as provide equal access to Covid-19 vaccines for undocumented migrants.
This is a good and timely paper providing necessary information and arguments to ensure the vaccination system open and user-friendly for undocumented migrants globally.
I suggest the authors to include some information about barriers and vaccination status of undocumented migrants some lower income countries (e.g. countries in Asia or Latin America) to give a comprehensive picture for all undocumented migrants at the global level.
Reviewer 4 Report
Dear Authors,
I honestly find your work pretty interesting although the scope is not completely clear to me. While in the first part you have done a detailed review of the country cases in terms of both, laws and policies towards migrants and Covid-19 vaccination programme, in the Conclusions you have concentrated the focus on the policies.
I think you should answer better (in the Introduction and Evolving picture sections) to the question: why do undocumented migrants are the most vulnerable groups? You miss some medical evidence on that.
For this purpose, you might be interested into reading some papers, among others, as:
- The political determinants of the health of undocumented immigrants (Piccoli and Wenner, 2022)
- Right of access to healthcare for undocumented migrants (Borges and Guidi)
- and other you can find in both Biblography
Round 2
Reviewer 3 Report
SUMMARY: Unfortunately, the paper still fails to address the topic with sufficient rigorousness and added value. Therefore, it still requires major revision to improve both content and form.
GENERAL CONCEPT
Consistency: there are still inconsistencies/mismatches across title, keywords, abstract, and actual scope of work and content.
· The paper covers several population groups (refugees, asylum seekers, displaced persons, documented and undocumented migrants). However, the title is about undocumented migrants only.
· The paper now covers both supply and demand sides barriers. However, “access barriers” (which relates to the supply side) is incorrectly referred to the demand side too.
Terminology: medical and epidemiological terminology still needs to be improved, beyond adopting the specific corrections that I provided in first review.
References: the entire set of references still needs to be improved and corrected.
· Key references: it is challenging to track and review revisions in references. However, the Introduction section is now more acceptable.
· Self-citations are still evident in the paper. Even one of the two references in the very opening line in the Introduction section is a self-citation. Furthermore, one self-citation is listed twice in the reference list (#2 and 121).
· Mismatches: there are still several mismatches between the text content and the cited references.
Methods and limitations: several issues with the newly added Methods and Limitations section.
· It needs to be moved up, as it typically follows an Introduction section and precedes a Results section. But this paper’s structure and content are problematic: the Introduction section goes beyond, as not only it is long but also covers some of the paper’s overall topic which is then contextualized in the Country case studies.
· It explains this paper is the actual proceedings of a meeting plus additions. If so, it should be clearly stated upfront in the title. Alternatively, the authors may write a meeting’s report and disseminate through channels other than a peer-reviewed journal.
· It does not justify the limited inclusion of evidence on the topic, which is becoming growingly available, also through primary data collection in the field. This type of evidence should make the core object of a review and would add value to it. The authors may research further and develop a review that is more evidence-informed. See comment from first review on Lines 146-150 and 169-173: they were pointing in the right direction raising interesting facts that needed to be better quantified/qualified to add value to the review but were eliminated.
· Data sources are not adequately explained. In particular, with regard to country-specific vs. general (non-country specific) observations: it is unclear if the country case studies provide the basis for the general (non-country specific) observations, which are currently in both the Introduction and Conclusions sections.
· Lines 217-219 do not seem to fit well in a Methods and Limitations section.
Writing style and length: The paper still reads quite prolix, and the file does not allow to track the reported cuts. In fact, the first file had 24 pages and the second file has 30 pages. Furthermore, there are several typos and syntax issues. Further streamlining and proofreading are needed.
REVIEW
CONTENT
Topic: the reported revisions do not seem to improve the paper’s intended focus on undocumented migrants.
· The paper does not compare population groups (as the Authors state in their revisions), but rather combines them. As a result, it is not possible to refer observations to one group only (undocumented migrants) as they refer to several population groups.
· The paper now covers both supply and demand sides barriers. However, “access barriers” (which relates to the supply side) is incorrectly referred to the demand side too.
Analysis and discussion: the review still dwells quite long on the overall COVID-19 pandemic and vaccination campaigns, rather than the vaccination campaign and undocumented migrants. The newly merged Introduction section reads like a synthesis of the paper’s topic, then contextualized in the country case studies. In fact, the paper’s structure doesn’t seem OK, and the reader gets lost. It may help to restructure, for example: introduction, methods and limitations, country case studies, analysis and discussion, conclusions. The same should be reflected in the Abstract. Please make sure that content fits in respective suggested sections, especially for general (non-country specific) text, which is sometime repeated across introduction and conclusions sections, and is not clear in relation with the country case studies (is it based on the country case studies or else?)
SPECIFIC COMMENTS
Abstract: Undocumented migrants are not displaced persons. The authors may stick to official definitions.
Keywords: The paper now covers both supply and demand sides barriers. However, “access barriers” (which relates to the supply side) is incorrectly referred to the demand side too.
Introduction section: “(…) past experiences and current situations (…)”: not meaningful in a health peer-reviewed journal.
Line 67: Overly detailed/technical, and inconsistent with the rest of the paper that doesn’t define units of measure even more directly related to the topic.
Line 108: “(…) use of vaccination in public health”: not a meaningful formulation in a health peer-reviewed journal.
Line 161: the suggested paper has been added in the reference list but key points could have been reflected in the review to help substantiate some of the general statements through findings of papers like this with primary data collection from the field.
Line 162: Still overly long and unclear.
Line 113: If this paper is the actual proceedings of a meeting plus additions, it should be clearly stated upfront in the title. Alternatively, the authors may write a meeting’s report and disseminate through channels other than a peer-reviewed journal.
Former Line 146-150 and 167-173: It is a pity these were just eliminated. This type of content should make the core object of a review and would add value to it. Those observations were pointing in the right direction raising interesting facts that needed to be better quantified/qualified to add value to the review, not eliminated. Also, the paper has been added in the reference list but key points could have been reflected in the review to help substantiate some of the general statements through findings of papers like this with primary data collection from the field.
COVID-19 vaccination and undocumented migrants section: There are recurring themes in the country case studies that could be presented and analyzed in a more systematic manner, as a quality feature of a “review”. The authors framed the review geographically (which is good) but could have unpacked it based on themes/issues. The country classification based on scorecard with criteria (Lines 530-532) can provide such approach and simplify the still long and somewhat redundant content.
Line 390: reference #85 is not matching with content.
Line 28: in this context, “vaccination” is more appropriate than “vaccines”. The authors may revise.
Line: the added value of this review would have been to describe in a systematic manner (not only mentioned), and advocate for these good practices to be adopted
Lines 782-785: unclear, does not make sense, and the references do not match.
